# Brief digital self-care intervention for health anxiety in a Swedish Medical University Clinic: a prospective single-group feasibility study

Susanna Österman  ,[1] Amira Hentati,[1] Erik Forsell,[1] Erland Axelsson  ,[2,3] Erik Hedman-Lagerlöf,[1,4] Nils Lindefors,[1] Volen Z Ivanov,[1] Martin Kraepelien[1]

**To cite:** Österman S, Hentati A, Forsell E, *et al.* Brief digital self-care intervention for health anxiety in a Swedish Medical University Clinic: a prospective single-group feasibility study. *BMJ Open* 2023;**13**:e077376. doi:10.1136/bmjopen-2023-077376

¹Department of Clinical Neuroscience, Karolinska Institutet, Stockholm, Sweden
²Division of Family Medicine and Primary Care, Department of Neurobiology, Care Sciences and Society, Karolinska Institutet, Stockholm, Sweden
³Liljeholmen University Primary Health Care Center, Academic Primary Health Care Center, Stockholm, Sweden
⁴Gustavsberg University Primary Care Center, Academic Primary Care Center, Stockholm, Sweden

**Correspondence to**
Susanna Österman;
susanna.osterman@ki.se

## ABSTRACT

**Objectives** In routine psychiatric care in Stockholm, Sweden, a comprehensive therapist-guided intervention for clinically significant health anxiety is implemented. However, there is a need for more easily accessible self-care interventions to improve treatment dissemination. This study aimed to transform an existing therapist-guided digital intervention into a self-care intervention, reducing patient burden and used clinical resources while maintaining quality and safety.

**Design** An uncontrolled feasibility study.

**Setting** Conducted at Karolinska Institutet, a medical university in Sweden, with nationwide recruitment trough online advertisements.

**Participants** Twenty-five adults used the self-care intervention and underwent telephone assessments, along with completing self-rated questionnaires.

**Intervention** The newly developed 8-week self-care intervention was designed to be user-friendly without therapist guidance, and to facilitate high levels of behavioural engagement.

**Primary and secondary outcome measures** Indicators of quality and safety, including changes in health anxiety severity (primary), clinician time, participant adherence, perceived credibility/satisfaction with the intervention and adverse events, were benchmarked against a previous study of the more comprehensive intervention it was based on.

**Results** Compared with the original guided intervention, the self-care intervention was condensed in terms of text (up to 70% less reading), duration (8 weeks instead of 12) and number of exercises. Quality indicators were similar to the original version. Most participants worked actively with core components in the self-care intervention. Within-group effects on health anxiety from pretreatment to the 3-month follow-up were large (g=1.37; 95% CI 0.74 to 2.00). No serious adverse events were reported.

**Conclusions** This brief digital self-care intervention shows potential for increasing access to treatment for individuals with health anxiety while reducing the burden on patients and clinical resources. Future studies should investigate the optimal type of intervention and support for different individuals, and if non-inferiority can be established.

**Trial registration number** NCT05446766.

## STRENGTHS AND LIMITATIONS OF THIS STUDY

⇒ Strengths of the study included the thorough diagnostic assessment and the low attrition rates at follow-up.
⇒ The results were benchmarked against a previously published randomised trial.
⇒ The lack of a control group as well as the small study sample limits the possibility to make causal inferences and calculate precise estimates of change.
⇒ All participants were self-referred and, hence, particularly motivated for treatment, which may have influenced the results.

## INTRODUCTION

Clinically significant health anxiety is a psychiatric condition characterised by an excessive fear of, or preoccupation with, having or developing a serious medical illness.[1] Individuals who suffer from clinically significant health anxiety tend to report high distress, disability and increased health service utilisation.[2] Therefore, the development of accessible, evidence-based interventions for this debilitating condition is a major challenge for healthcare today.

Clinically significant health anxiety can be effectively treated with therapist-guided internet-based cognitive behavioural therapy (ICBT),[3] a format which requires less therapist support per patient compared with traditional face-to-face cognitive behavioural therapy (CBT), thus making the treatment more cost-effective.[4–6] In order to further increase the clinical utility of CBT, the development of programmes without therapist support, that is, self-guided or self-care interventions, represent an attractive avenue because of their more immediate scalability. Self-care interventions may also reduce stigma, thereby decreasing barriers to seeking treatment.[7] Furthermore, data indicate that many individuals with severe health anxiety

**BMJ**

would prefer working with self-guided interventions over guided versions.[8]

A recurrent finding in research highlights the challenge of maintaining comparable quality in self-help interventions, like that of guided interventions.[9] However, there are promising studies suggesting that self-care interventions can be developed to a level where they can be comparable to guided ones.[10] Several features have been cited as important in order to maintain effectiveness when clinician guidance is removed.[11] These include a clinical interview prior to and post-intervention, monitoring of participant progress, high-quality treatment material and automated features such as automatic messages prompting and reinforcing participants' work with the intervention.

In Sweden, a 12-week guided exposure-based digital intervention for health anxiety[12–14] has been successfully implemented in routine psychiatric care at the Stockholm Internet psychiatry unit since 2018 (https://www.internetpsykiatri.se/en/). Exposure-based therapy, comprising the key component exposure with response prevention (ERP), involves gradual exposure to feared health-related situations or stimuli while refraining from compulsive health-related behaviours.[15 16]

To further meet patient needs, enhance treatment accessibility, and reduce the time and costs associated with clinician support, this comprehensive digital intervention could potentially be offered as a self-help intervention. The guided 12-week digital intervention has, in addition to being used in a guided format, previously been used self-guided but otherwise unaltered version.[14] This self-guided version showed expected and clinically significant effects but somewhat lower adherence compared with the guided intervention (around 50% completed modules compared with 70%). In both the guided and self-guided versions of the intervention, the key component ERP was not introduced before weeks 4–5 in the treatment programme. This means that many individuals with low treatment adherence, especially in the unguided self-guided version, did not have sufficient time to work with this key component. Other drawbacks of the original intervention include its relative comprehensiveness, which places significant demands on the patient, and the lack of optimisation in terms of user-friendliness or suitability for self-guided use.

The development of a brief self-care intervention with the goal of increasing active engagement with the main treatment components and delivering it within a structured care process, may have the potential to improve its scalability. It could decrease participant burden and clinician resources, while maintaining the inherent quality seen in therapist-guided interventions. However, before conducting a large, randomised controlled trial, an evaluation of issues related to the feasibility and acceptability of such an intervention should be conducted.

The aim of this study was, thus, to evaluate whether individuals with health anxiety find the intervention acceptable, adhere to it over time, and if it can be safely used without therapist guidance. Additionally, we aimed to determine how much clinician time such an intervention would require. Feasibility measures included participant adherence, perceived treatment credibility/satisfaction, adverse effects and clinician time. Finally, we aimed to evaluate the preliminary efficacy of this brief intervention on symptoms of health anxiety. Measures of feasibility as well as preliminary efficacy were benchmarked to a previous study based on the more extensive and resource-intensive therapist-guided intervention it was based on.[14]

## MATERIAL AND METHODS
### Context and design
This study was set up within the framework of a digital self-care development project initiated in 2018 and funded by the Swedish Ministry of Health and Social Affairs. The primary goal of this project is to improve the accessibility of psychological treatment in Sweden by creating concise digital self-care interventions derived from already existing digital and comprehensive therapist-guided interventions. The prospective single-group feasibility study was conducted at the Centre for Psychiatry Research, Stockholm Health Care Services and Karolinska Institutet, Sweden. The study employed nationwide recruitment through advertisements on social media platforms. The study was preregistered on ClinicalTrials.gov (NCT05446766). Measures of feasibility and preliminary effects were benchmarked against a more comprehensive digital intervention, in both guided and unguided versions from a previously published randomised trial.[14]

### Patient and public involvement
Patients were not involved in the design of the study or the intervention. However, the research team possesses significant clinical experience with the patient group as all are clinicians/researchers. Moreover, participants underwent a telephone interview after the intervention, during which they were asked about their experience with the intervention. These comments will be used in future revisions of the intervention.

### Participants and recruitment
For evaluation of the feasibility measures, a sample of 20 participants was deemed to be sufficient. Regarding preliminary within-group effect measured with the 14-item Short Health Anxiety Inventory (SHAI-14), to detect a moderate effect size (Hedges'g=0.6), a sample of 24 participants was required, given 80% power and an $\alpha$ level set to $p<0.05$ (two tailed). To allow for some attrition, the goal was to recruit 25–30 participants. These were recruited through advertisements in social media under the heading 'Do you worry a lot about your health?' and started the intervention after a clinical interview via telephone (average time: 40 min, SD: 6.9). This assessment was conducted by a clinical psychologist, and included the Health Preoccupation Diagnostic Interview (HPDI)[17] and the Mini-International Neuropsychiatric

Interview.[18] Participants also underwent an interview via telephone postintervention (average time: 23 min, SD: 5.7). Inclusion criteria were (1) age of 18 years or older, (2) a principal diagnosis of illness anxiety disorder or somatic symptom disorder (the prototypical diagnoses for clinically significant health anxiety according to the Diagnostic and Statistical Manual of Mental Disorders (DSM-5)),[1 19] (3) currently not receiving similar psychological treatment for health anxiety and (4) no serious medical illness or urgent need for more intensive psychiatric care. If participants were on psychotropic medication, the dose had to be stable for 4 weeks prior to the baseline assessment. Recruitment started on 6 September 2022 and the last follow-up was collected on 29 March 2023.

## Intervention

The digital intervention was based on the theoretical literature regarding self-help interventions as well as clinical experience with the patient group. The graphical user interface was developed together with experts on user experience, with the aim of making the intervention easy to use without therapist guidance and to facilitate high levels of behavioural engagement.[20] The self-care intervention was provided within a structured care process that included clinical interviews before and after the intervention. These interviews were considered a vital part of the intervention, ensuring diagnostic accuracy and was intended to be a motivational tool, encouraging participants to adhere to the intervention components. Participants also received automated text messages in order to encourage engagement with the treatment. If in need of technical support, they could ask for assistance via the platform and were then contacted via telephone or email within approximately 1–3 days. For a flow chart of the clinical procedures, see figure 1.

The content of the intervention was based on an already existing guided digital intervention for health anxiety but was adapted to be self-guided. One of the main goals was to make the intervention less demanding for the participant. This was achieved by changing aspects such as treatment length, the number of therapeutic techniques and the complexity of the treatment material. See table 1 for a detailed comparison between the new self-care intervention and the original comprehensive digital intervention in both therapist-guided and self-guided versions. In the new intervention, the primary therapeutic focus remained on ERP,[16] a highly demanding therapeutic technique. To alleviate participant burden, extraneous elements such as mindfulness training were excluded. Mindfulness was viewed not as a standalone technique but as a tool for exposure facilitation in the original intervention. In the new self-care programme, ERP adherence was bolstered through early presentation, weekly problem-solving sections, clinical vignettes guiding ERP challenges and an optimised user interface. These enhancements aimed to boost participant engagement with the central ERP component.

ERP was introduced in the very first week of the intervention. ERP is thought to decrease symptoms of health anxiety through repeated confrontation with feared or avoided stimuli (eg, body sensations, situations) while refraining from engaging in maladaptive health-related behaviours (eg, body checking).[15 16] The intervention included both mandatory and optional text sections with the latter mainly consisting of different clinical vignettes in order to make the intervention more personalised. It also included illustrative pictures to enhance the clarity of the content, as well as expandable 'learn more' options. These features were designed to limit the amount of content displayed simultaneously and make it less demanding for the participant to absorb the material. They were also aimed to help patients differentiate between what we considered 'necessary psychoeducation' and 'extracurricular' texts for those who wanted more. See figure 2 for comparative screenshots.

## Measures

The HPDI[17] is a structured interview designed to aid the assessment of illness anxiety disorder and somatic symptom disorder according to DSM-5 criteria. The HPDI can be administered with high inter-rater reliability and was administered via telephone pretreatment and at 12-week follow-up.[17]

Self-report questionnaires were administered on the same technical platform where the self-care intervention was delivered, and participants received text reminders about these. The primary outcome was a change in health anxiety as measured with the widely used SHAI-14, which has excellent psychometric properties[21] and shows evidence of good diagnostic utility in both clinical and non-clinical settings.[22] Participants completed the SHAI-14 at baseline, each week during treatment, after treatment (week 8), after four additional weeks (week 12) and at the 3-month follow-up (week 24).

To assess credibility and expectancy, the five-item version of Credibility/Expectancy Questionnaire was administered 2 weeks after treatment start.[23] A total score of 30 points or higher is considered to indicate adequate treatment credibility. At post-treatment, satisfaction with the treatment was evaluated using the Client Satisfaction Questionnaire-8.[24] To assess safety, information on possible adverse events was collected with an open-ended question at post-treatment where participants reported whether they had experienced

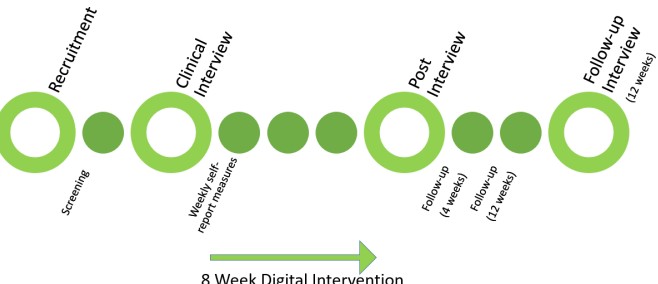

**Figure 1** Flow chart of the clinical procedures.

**Table 1** Overview of the themes of the original and the new interventions including word count

| Week | Original digital intervention (same content in guided and self-guided version) Content | Word length (n) | New digital self-care intervention Content | Word length (n) Mandatory | Word length (n) Optional |
|------|------|------|------|------|------|
| 1 | Introduction to CBT and health anxiety. How to navigate the programme. Introduction to mindfulness and behaviour diary. | 8528 | Information about health anxiety. How to navigate the programme. Psychoeducation about ERP | 5857 | 2493 |
| 2 | Presentation of the CBT model of severe health anxiety, continued mindfulness training and behaviour diary. | 6558 | The role of bodily symptoms. Exposure: interoceptive exposure. | 2564 | 1248 |
| 3 | Cognitive processes, continued mindfulness training. | 7699 | The role of safety behaviours and the importance of response prevention. Participants continue to practice ERP. | 2878 | 3185 |
| 4 | Introduction to exposure: interoceptive exposure exercises, continued mindfulness training | 7961 | Avoidance behaviour and exposure to health anxiety provoking stimuli (in vivo exposure). Participants continue to practice ERP | 1610 | 3059 |
| 5 | Response prevention, continued mindfulness training. Participants continue to practice ERP | 7454 | Thoughts: Exposure to illness thoughts through imaginal exposure. Participants continue to practice ERP | 2120 | 4730 |
| 6 | Exposure to health anxiety provoking stimuli (exposure in vivo). Participants continue to practice ERP | 5702 | Thoughts: Exposure to illness thoughts through imaginal exposure. Participants continue to practice ERP | 1418 | 4404 |
| 7 | Imaginal exposure and exposure to the fear of death. Participants continue to practice ERP | 5381 | Exposure and quality of life. Participants continue to practice ERP | 1555 | 3748 |
| 8 | Continued imaginal exposure. Participants continue to practice ERP | 3486 | Relapse prevention: construction of a relapse plan and plan for future goals. How to handle healthcare utilisation in the future. | 1845 | 2066 |
| 9 | Common obstacles to exposure. Participants continue to practice ERP | 1504 | NA | | |
| 10 | Setting new goals. Participants continue to practice ERP | 396 | NA | | |
| 11 | A summary of the treatment. Values and quality of life. Participants continue to practice ERP | 3197 | NA | | |
| 12 | Maintaining gains and preventing relapse. How to handle healthcare utilisation in the future. | 3522 | NA | | |
| Total | | 61 388 | | 19 847 | 24 933 |

Original digital intervention refers to the intervention described in Hedman et al.[14]
CBT, cognitive–behavioural therapy; ERP, exposure with response prevention; NA, not applicable.

any negative or unwanted effects of the treatment. Adherence was determined in terms of the number of completed modules, and through weekly questions about use of ERP. Additional measurements of psychiatric symptoms, quality of life and adherence are described in online supplemental file 1.

### Analysis

Most quality indicators, including adherence, patient feedback and symptom outcomes, were reported descriptively. All analyses were based on observed data. Within-group standardised effect sizes were calculated for symptom change (Hedges' g based on pooled SD with

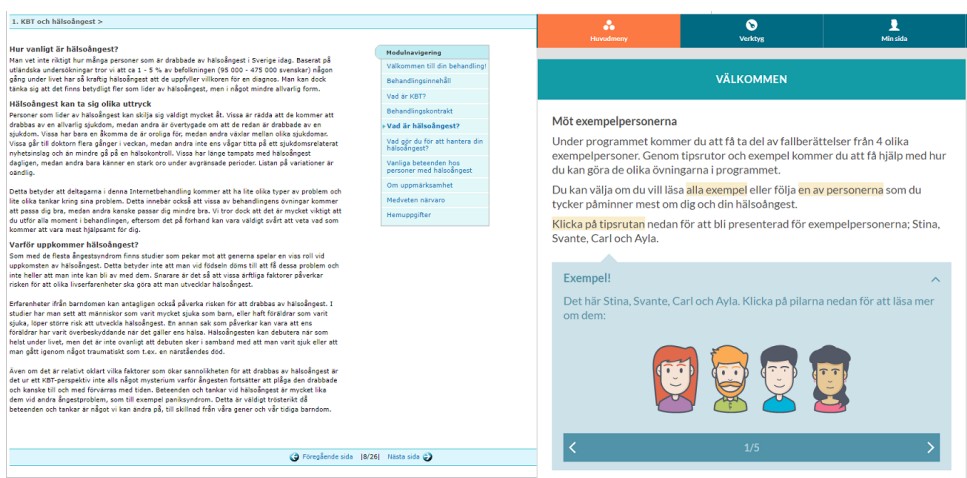

**Figure 2** Screenshots from the original intervention (left) and the new self-care intervention (right).

95% CI) at week 12 and at the 3 months follow-up. Week 12 was chosen to be able to compare the within-group effect sizes with the 12-week long interventions in the reference study.[14] This comparison was with previously published data and was intended as a non-randomised benchmark. Remission was defined as scoring below 18 points on the SHAI-14.[22]

## RESULTS

In total, 25 out of 33 interviewed participants were included and received access to the self-care intervention. Twenty-two participants (88%) provided outcome data at the week 12 follow-up, and 23 (92%) did so at the 3-month follow-up. Overall, participant demographics were comparable to those in the reference study. For participant demographics, revisions of the intervention and comparisons of quality indicators between the new self-care intervention and the reference study (see table 2). For an overview of the proportions of participants who scored below 18 points on the SHAI-14 (criteria for remission) between time points (see figure 3). All participants except two (23/25, 92%) reported having worked with ERP during the treatment period. Regarding safety, three participants of those who answered the questions on adverse events (3/20, 15%) experienced an adverse event during the treatment period. These were (1) increased anxiety, (2) increased awareness of bodily symptoms and (3) feelings of being alone/exposed and not being able to discuss this with a therapist. There were no reports of serious adverse events. See online supplemental file 1 for outcomes on additional secondary measures.

## DISCUSSION

The aim of this study was to develop and evaluate the feasibility of a newly developed brief digital self-care intervention for individuals with clinically significant health anxiety and compare it to the more comprehensive and resource-demanding digital intervention it was based on. The purpose of this was to further enhance treatment accessibility for a patient population that currently presents a challenge to the healthcare system. In comparison to both the therapist-guided and self-guided version of the original intervention in the reference study, quality indicators such as participant adherence, satisfaction with treatment, preliminary within-group effect sizes as well as safety were promising. Importantly, almost all participants in the new self-care intervention worked with ERP during the treatment period.

Compared with the reference study, the participants appear to share similar demographics and disease burden and the time spent on interviews of around 60 min per participant was also similar.[14] However, in the new self-care intervention, the overall clinician workload was reduced since no written guidance were provided which was the case in the guided original intervention where clinician on average posted 18.4 text messages (SD 9.6), mean time: 63.6 min (SD: 35.4). Moreover, in both the guided and the self-guided original intervention, the clinician contacted the participant via SMS or phone if they had not been active on the platform for a week, which was not the case in the new self-care intervention. This moderately high level of structure and contact may have worked as a form of engagement enhancing procedures—similar to written guidance. The relatively high level of adherence observed in the new self-care intervention, despite the absence of such contact, is therefore promising and aligns with previous studies indicating that a well-designed self-help intervention may compensate for the lack of therapist support.[11 25] Furthermore, preliminary within-group effects seemed promising given the fact that the new self-care intervention posed less burden on participants due to a shorter treatment length, less and more simplified text and a significant reduction in the number of different types of mandatory exercises. Since the guided intervention in the reference study had seemingly higher remission rates at the follow-up assessment, there may, however, be some participants who would benefit from receiving additional guidance. Future studies could help identify who would benefit from such guidance.

**Table 2** Participant demographics, programme characteristics and quality indicators benchmarked against the original interventions in the reference study

| Part of treatment | Original digital intervention (guided version) | Original digital intervention (self-guided version) | New digital self-care intervention |
|---|---|---|---|
| Demographics | | | |
| | (n=32) | (n=33) | (n=25) |
| Gender, n (%) | 22 (88%) female | 24 (73%) female | 21 (84%) female |
| Age | | | |
| Mean age (SD) | 38.6 (12.6) | 37.4 (11.6) | 41.0 (15.2) |
| Minimum-maximum | 22–70 | 20–72 | 21–70 |
| Occupational status, n (%) | | | |
| Working full time/part time | 22 (69%) | 27 (82%) | 15 (60%) |
| Retired, on sick-leave or unemployed | 3 (9%) | 3 (9%) | 7 (28%) |
| Other | 7 (22%) | 4 (12%) | 8 (32%) |
| Marital status | | | |
| Married or de facto, n (%) | 28 (88%) | 29 (88%) | 17 (68%) |
| Years with severe health anxiety mean (SD) | 7.2 (5.7) | 7.8 (6.4) | 9.0 (9.4) |
| Psychiatric comorbidity* | | | |
| Depressive disorder | 1 (3%) | 11 (33%) | 5 (20%) |
| Anxiety disorder | 14 (44%) | 16 (48%) | 13 (52%) |
| Stabilised psychotropic medication, SSRIs or SNRIs, n (%) | 7 (22%) | 7 (21%) | 8 (32%) |
| Severity at pre SHAI-14 (0–42) | 26.8 (5.4) | 28.2 (6.1) | 25.7 (6.1) |
| Programme | | | |
| Duration | 12 weeks | 12 weeks | 8 weeks |
| User interface | Desktop optimised | Desktop optimised | Mobile optimised |
| No of words | 60 426 | 60 426 | Mandatory: 19 847 Optional: 24 933 |
| Therapist time | M (SD)=63.6 (35.4) min/participant | 0 min by design | 0 min by design |
| Means of support | Personalised therapist messages | None | Generalised automated text messages |
| Inactivity reminders | Text-message or phone call from clinician | Text-message or phone call from clinician | Generalised automated text messages |
| Unlocking new modules | Unlocked from start with instruction to complete them in consecutive order | Unlocked from start with instruction to complete them in consecutive order | Automated if/when: 1 week had passed+participant interacted with all content |
| Quality indicators | | | |
| Adherence, modules M (SD), % | 8.6 (3.0), 72% | 6.6 (3.6), 55% | 5.3 (2.8), 66% |
| Treatment credibility M (SD) | 37.1 (6.9) | 36.8 (7.9) | 33.5 (13.4) |
| Completed 12-week quality indicators, n (%) | 31 (97%) | 32 (97%) | 22 (88%) |
| Effect size (95% CI) Within-group effect on primary outcome 12 weeks† | 1.55 (0.87 to 2.22) | 1.31 (0.84 to 1.78) | 1.42 (0.78 to 2.06) |
| Below cut-off (remission) at 12 weeks†, n (%, 95% CI) | 19, (61%, 95% CI=42% to 78%) | 17 (53%, 95% CI=35% to 71%) | 14 (64%, 95% CI=41% to 83%) |
| Treatment satisfaction M (SD) | 26.4 (4.6) | 24.8 (4.1) | 23.6 (5.3) |
| Adverse events (at least one) n (%) | 6 (19%) | 4 (12%) | 3 (14%)‡ |

Continued

Österman S, *et al. BMJ Open* 2023;**13**:e077376. doi:10.1136/bmjopen-2023-077376

**Table 2** Continued

| Part of treatment | Original digital intervention (guided version) | Original digital intervention (self-guided version) | New digital self-care intervention |
|---|---|---|---|
| Completed FU quality indicators, n (%) | 31 (97%) | 27 (82%) | 23 (92%) |
| Effect size (95% CI) Within-group effect on primary outcome FU§ | 2.23 (1.31 to 3.14) | 1.52 (0.90 to 2.15) | 1.37 (0.74 to 2.00) |
| Below cut-off Remission at FU§, n (%, 95% CI) | 20 (65%, 95% CI=45% to 81%) | 17 (63%, 95% CI=42% to 81%) | 12 (52%, 95% CI=28% to 72%) |

Original digital intervention refers to the intervention described in Hedman *et al*.[14]
*Psychiatric comorbidity refers to having a current DSM-5 diagnosis according to the MINI.[18] 'Depressive disorder' refers to major depressive disorder. 'Anxiety disorder' refers to panic disorder, agoraphobia without panic disorder, social anxiety disorder, obsessive–compulsive disorder, generalised anxiety disorder and post-traumatic stress disorder.
†12 weeks refers to post-treatment in the reference study and 4 weeks after post-treatment in the new intervention.
‡Adverse events were reported at post-treatment in this study (week 8), number of participants who completed the measurement was n=20.
§FU period refers to 6 months after post-treatment in the reference study and 3 months after post-treatment in the new intervention. Remission is defined as a score below 18 points on primary outcome.
DSM-5, The Diagnostic and Statistical Manual of Mental Disorders, Fifth Edition; FU, follow up; M, mean; MINI, Mini-International Neuropsychiatric Interview; SHAI-14, 14-item Short Health Anxiety Inventory; SNRIs, serotonin and norepinephrine reuptake inhibitors; SSRIs, selective serotonin reuptake inhibitors.

Although preliminary, this study supports the notion that a brief digital self-care intervention is feasible and can be comparable to more comprehensive guided versions and should be further examined in larger studies. In order to maintain adherence as well as treatment effects, factors such as the self-care intervention being provided within a clinical context with a short clinical interview prior and post to the intervention are potentially important.

The study has some limitations such as its small sample size and the absence of randomisation. Since the comparisons with the original comprehensive intervention in the reference study were not randomised, there were some apparent differences in population characteristics, such as

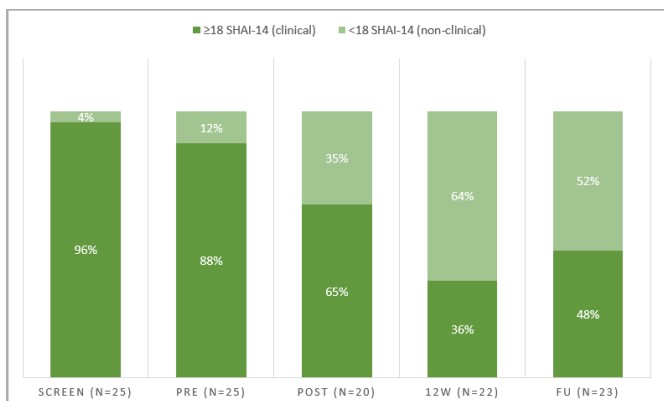

**Figure 3** Proportion scoring below/above the cut-off for remission proportions of participants scoring below/above the cut-off for remission (18 points) on the SHAI-14 at different time points throughout the trial. Post refers to 8 weeks after treatment start. Twelve weeks refers to 4 weeks after post-treatment (week 12). FU refers to the 3-month FU (week 24). FU, follow-up; SHAI-14, 14-item Short Health Anxiety Inventory.

levels of psychiatric comorbidity. Additionally, the recruitment methods differed between the reference study and the current study. The reference study used both advertisements and recruitment from healthcare professionals, while the current study solely relied on advertisements. This variation may have caused sample disparities. However, only 31 out of 132 participants in the reference study were recruited from a medical clinic. Further, the period between the two studies included a pandemic, potentially influencing the population's mental health awareness and levels of worry. This situation might have attracted different types of patients interested in participating in an intervention focused on health anxiety. These factors render caution in all interpretation of differences and similarities in the comparison of interventions. Nevertheless, a strength lies in the fact that all participants in both the original and current studies underwent the same thorough diagnostic assessment, the two studies use the same eligibility criteria as well as that the level of health anxiety measured with a well-validated measure appears to be similar. The current study primarily employed self-report measures. It is important to bear in mind the limitations inherent in these types of instruments, such as the reliance on the cooperation of the person answering them and the potential presence of recall and response bias. A potential improvement in a future study could be to conduct diagnostic interviews at post-treatment by a blind assessor and/or measure changes in health-care consumption before and after treatment, based on patient register data. Further, many adjustments from the original intervention were made, both regarding content and intervention processes. These changes have made it difficult to evaluate the feasibility of each component separately. Finally, the sample consisted of self-recruited

volunteers. However, if implemented, a brief self-care intervention such as the current one may be a means to reach individuals who prefer working on their own and/or who would not otherwise seek help due to stigma or other reasons. It could also possibly serve as a first low-intensity intervention in a stepped care model. Additionally, the vast majority of patients receiving ICBT in implemented regular care settings are also self-referred.[26]

In conclusion, the quality indicators did not seem to show any substantial differences between the brief self-care intervention compared with the original more resource demanding intervention. This highlights the need for larger randomised studies evaluating what type of intervention works for whom, the role of guidance as well as the level of comprehensiveness from both a cost-effectiveness as well as participant and clinician burden perspective, as well as studies of possible non-inferiority. Future research should investigate the impact of different treatment components on outcomes, as well as participants' satisfaction and adherence. A study using a factorial design,[27] adjusting treatment components, therapist support levels and treatment duration, could shed light on these aspects. If this brief self-care intervention is found to be both effective and resource-effective, it could be a highly useful treatment option for patients with clinically significant health anxiety.

**Contributors** SÖ: study design, intervention design, drafting of manuscript, statistical analysis and conducting clinical interviews. MK: study concept, intervention design, study design, principal investigator. AH: conducting clinical interviews, contributions to manuscript. EF: Study concept, intervention design, study design. EA: statistical analysis and contributions to manuscript. EH-L: study concept, contributions to manuscript. NL and VZI: study concept, study design, contributions to manuscript. Guarantor: SÖ. All authors: critical review and revision of manuscript.

**Funding** This study was supported by the Swedish Ministry of Health and Social affairs (grant number S2018/03855/FS).

**Competing interests** EH-L and EA have coauthored a self-help book for pathological health anxiety centred around exposure-based cognitive behaviour therapy, and for which they receive royalties.

**Patient and public involvement** Patients and/or the public were not involved in the design, or conduct, or reporting, or dissemination plans of this research.

**Patient consent for publication** Not applicable.

**Ethics approval** The new self-care intervention was developed based on a previously evaluated treatment protocol which has been deemed safe. Study registration was administered online on a secure platform where prospective participants provided digital informed consent and contact information. All individuals who registered to the study were contacted and informed of whether they qualified to participate. The authors declare that all procedures contributing to this work comply with the ethical standards of the relevant national and institutional committees on human experimentation and with the Declaration of Helsinki of 1975, as revised in 2008. All procedures involving human subjects/patients were approved by the Swedish Ethical Review Authority (identifier 2021-04133).

**Provenance and peer review** Not commissioned; externally peer reviewed.

**Data availability statement** Data are available on reasonable request.

of the translations (including but not limited to local regulations, clinical guidelines, terminology, drug names and drug dosages), and is not responsible for any error and/or omissions arising from translation and adaptation or otherwise.

**ORCID iDs**
Susanna Österman http://orcid.org/0000-0001-5031-5025
Erland Axelsson http://orcid.org/0000-0003-2562-2925

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
