## [Reviewer comments · BMJ Open]

ARTICLE DETAILS

TITLE (PROVISIONAL)	A Brief Digital Self-Care Intervention for Health Anxiety in a Swedish Medical University Clinic: A Prospective Single-Group Feasibility Study
AUTHORS	Österman, Susanna; Hentati, Amira; Forsell, Erik; Axelsson, Erland; Hedman-Lagerlöf, Erik; Lindefors, Nils; Ivanov, Volen; Kraepelien, Martin

VERSION 1 – REVIEW

REVIEWER	Cross, Shane P Orygen Ltd
REVIEW RETURNED	20-Aug-2023

GENERAL COMMENTS	he self-care intervention being provided within a clinical context with a short clinical interview prior and post to the intervention are potentially important.
--

REVIEWER	Irmansyah, Irmansyah National Research and Innovation Agency Republic of Indonesia, Health Research
REVIEW RETURNED	22-Aug-2023

GENERAL COMMENTS	The study addresses a relevant and important topic in the field of mental health, specifically on health anxiety, which is a common clinical problem that often goes undetected and untreated. The study aims to evaluate the efficacy of a low-intensity CBT intervention that may fill the gap in the access of intervention. The use of multiple outcome measures in line with a comparison design with the previously original version research outcome is a good and effective method to examine the effectiveness of the new short version of the unguided digital intervention. The results show the non-inferiority of the new intervention, which provides a good foundation for further more rigorous and robust research design. There is room to improve the manuscript. Below are some of my suggestions: • Please give a brief description of the development of the new version. How did the researchers come up with the idea and the process to develop the new short version? Was there any patient involvement in this process? It was mentioned in the manuscript that patients were not involved in the research design or the intervention, but the involvement of patients and public can also be in other aspects of the research process/procedure.• Please mention that, in addition to reducing the words, pictures were also inserted in the new version, which was a good approach to make it more simple but increase the clarity of the module. Furthermore, from table 1, it can be seen that the content of the
---

	intervention was also changed. For example, the mindfulness training, which was an important component of the original version, was removed. In the main reference (Hedman, E., 2016), the usefulness of mindfulness training in the module was described. Please provide a reason behind this change and whether it had any effect on the new short version of the intervention.  • Please elaborate more on the limitations of the study. Especially, when comparing with the original research that the intervention was based on, there are some other differences in the sample, besides the recruitment method that was mentioned. There is the time lapse, which was about 7 years, and there was a pandemic, which affected the population's awareness of mental health, which could bring different types of patients. Moreover, the original research recruited more general population, while in this study, the sample was recruited from a medical academic community. • All of the outcome measures were mostly self-report and lacked clinical validation. However, self-report measures may be subject to some biases, such as response bias and recall bias, and may not reflect the actual clinical diagnosis or severity of the disorders. How can these limitations be address or mitigate in the future studies?
--	--

REVIEWER	Miller, J Washington University, Biostatistics
REVIEW RETURNED	16-Oct-2023

GENERAL COMMENTS	This is an informative case report which provides useful information about self administered CBT computer interventions. The methods suggest that the sample size was chosen to provide adequate precision in outcome measure estimates, but this is not defined quantitatively, e.g. a declaration about the widths of expected confidence intervals. Confidence intervals should be provided for binary outcome measures, e.g. remission. A list of the psychiatric comorbidities should be provided for the three groups of interest.
---

VERSION 1 – AUTHOR RESPONSE

Reviewer 1 comment #1:

“The self-care intervention being provided within a clinical context with a short clinical interview prior and post to the intervention are potentially important.”

We respond:

We thank the reviewer for this comment.

Reviewer 2 comment #1:

The study addresses a relevant and important topic in the field of mental health, specifically on health anxiety, which is a common clinical problem that often goes undetected and untreated. The study aims to evaluate the efficacy of a low-intensity CBT intervention that may fill the gap in the access of intervention. The use of multiple outcome measures in line with a comparison design with the previously original version research outcome is a good and effective method to examine the

effectiveness of the new short version of the unguided digital intervention. The results show the non-inferiority of the new intervention, which provides a good foundation for further more rigorous and robust research design.

We respond:

We thank the reviewer for this positive feedback.

Reviewer 2 comment #2:

Please give a brief description of the development of the new version. How did the researchers come up with the idea and the process to develop the new short version? Was there any patient involvement in this process? It was mentioned in the manuscript that patients were not involved in the research design or the intervention, but the involvement of patients and public can also be in other aspects of the research process/procedure.

We respond:

We thank the reviewer for this valuable comment. We have now elaborated on the information regarding why the intervention was developed and the work process. We have made changes to the manuscript in several sections:

Under the section "Context and design"

"This study was set up within the framework of a digital self-care development project initiated in 2018 and funded by the Swedish Ministry of Health and Social Affairs. The primary goal of this project is to improve the accessibility of psychological treatment in Sweden by creating concise digital self-care interventions derived from already existing digital and comprehensive therapist-guided interventions."

Under the section "Patient and Public Involvement"

"Patients were not involved in the design of the study or the intervention. However, the research team possesses significant clinical experience with the patient group as all are clinicians/researchers. Moreover, participants underwent a telephone interview after the intervention, during which they were asked about their experience with the intervention. These comments will be utilized in future revisions of the intervention."

And in the "intervention" section:

"The digital intervention was based on the theoretical literature regarding self-help interventions as well as clinical experience with the patient group. The graphical user interface was developed together with experts on user experience, with the aim of making the intervention easy to use without therapist guidance and to facilitate high levels of behavioural engagement."

"The self-care intervention was provided within a structured care process that included clinical interviews before and after the intervention. These interviews were considered a vital part of the intervention, ensuring diagnostic accuracy, and was intended to be a motivational tool, encouraging participants to adhere to the intervention components."

Reviewer 2 comment #3:

Please mention that, in addition to reducing the words, pictures were also inserted in the new version, which was a good approach to make it more simple but increase the clarity of the module.

We respond:

We thank the reviewer for this helpful comment. We have now made changes in the "intervention"-section:

"It also included illustrative pictures to enhance the clarity of the content, as well as expandable 'learn

more' options. These features were designed to limit the amount of content displayed simultaneously and make it less demanding for the participant to absorb the material. They were also aimed to help patients differentiate between what we considered "necessary psychoeducation" and "extracurricular" texts for those who wanted more."

Reviewer 2 comment #4:

Furthermore, from table 1, it can be seen that the content of the intervention was also changed. For example, the mindfulness training, which was an important component of the original version, was removed. In the main reference (Hedman, E., 2016), the usefulness of mindfulness training in the module was described. Please provide a reason behind this change and whether it had any effect on the new short version of the intervention.

We respond:

We thank the reviewer for this comment. The aim of the present study was to develop a brief intervention, with reduced treatment duration, briefer and more concise content, fewer and more focused treatment components, and different exercises. One of the main goals was to simplify the intervention and make it less demanding for participants to work with independently. One concern we thought of regarding having numerous components in the treatment was to confuse or overwhelm participants when working with the intervention in a self-guided format. For example, participants might work with several different components, but not well enough with any to yield positive effects from them. For this reason, only one component of the original treatment package was chosen. Since exposure with response prevention (ERP) was a corner stone of the original intervention, ERP was chosen as the main component of this new, brief intervention. Although mindfulness training and cognitive techniques were part of the original intervention, they were not standalone or main components. Instead, they served as facilitators, enhancing the likelihood that participants would engage in anxiety-provoking exposure exercises.

When developing the current brief intervention, our goal was to assist participants in conducting exposure exercises in various ways without requiring additional practices such as daily mindfulness training. This involved presenting ERP in the first week of treatment, providing weekly sections with examples of problems and suggested solutions, and offering clinical vignettes with instructions on how to stay present during ERP sessions. Additionally, the presentation of the intervention was collaboratively developed with experts on user experience, aiming to enhance participants' engagement with the main component. Finally, to reduce the intervention's demands, we shortened its length, minimized its text, and reduced the number of mandatory homework assignments. Due to the study's design, we cannot draw definitive conclusions about the efficacy of this new intervention. Nevertheless, the results suggest that the modified intervention is feasible and has yielded comparable ratings in terms of acceptability, adherence, and negative effects, which is encouraging. However, future studies should explore the relative importance of different treatment components in treatment effects, as well as participants' satisfaction and adherence. This could be achieved through a study using a factorial design, manipulating various treatment components, therapist support levels, and treatment duration.

We have made changes in the manuscript in several sections:

In the "intervention"-section:

"The content of the intervention was based on an already existing guided digital intervention for health anxiety but was adapted to be self-guided. One of the main goals was to make the intervention less demanding for the participant. This was achieved by changing aspects such as treatment length, the number of therapeutic techniques, and the complexity of the treatment material. See Table 1 for a detailed comparison between the new self-care intervention and the original comprehensive digital intervention in both therapist-guided and self-guided versions. In the new intervention, the primary therapeutic focus remained on ERP, a highly demanding therapeutic technique. To alleviate participant burden, extraneous elements like mindfulness training were excluded. Mindfulness was viewed not as a standalone technique but as a tool for exposure facilitation in the original intervention. In the new self-care program, ERP adherence was bolstered through early presentation, weekly problem-solving sections, clinical vignettes guiding ERP challenges, and an optimized user interface.

These enhancements aimed to boost participant engagement with the central ERP component."

In the "discussion" section

"Also, many adjustments from the original intervention were made, both regarding content and intervention processes. These changes have made it difficult to evaluate the feasibility of each component separately."

In the "discussion" section

"Future research should investigate the impact of different treatment components on outcomes, as well as participants' satisfaction and adherence. A study utilizing a factorial design, adjusting treatment components, therapist support levels, and treatment duration, could shed light on these aspects".

Reviewer 2 comment #5:

Please elaborate more on the limitations of the study. Especially, when comparing with the original research that the intervention was based on, there are some other differences in the sample, besides the recruitment method that was mentioned. There is the time lapse, which was about 7 years, and there was a pandemic, which affected the population's awareness of mental health, which could bring different types of patients. Moreover, the original research recruited more general population, while in this study, the sample was recruited from a medical academic community.

We respond:

We thank the reviewer for these comments and important reflections. We acknowledge the potential differences in the sample attributed to various factors such as time and recruitment methods, which may have influenced the composition of the sample. The original study employed newspaper advertisements and recruitment through mental health and primary care personnel, while the current study only used online advertisements. However, it's important to note that only a minority (31 out of 132) of participants in the original study were recruited from a medical clinic and the rest were self-referred from the general population. Further, we believe that the overall settings are quite similar; both studies were conducted at Karolinska Institutet with nationwide recruitment in Sweden with the aim to recruit from the general population. Moreover, both studies adhered to the same eligibility criteria and used the same structured diagnostic assessments.

Nevertheless, we agree that the distinct advertising methods and the temporal gap might have influenced the sample in ways that make direct comparisons even more challenging, a point that should be emphasized in the limitations section. Additionally, the description of the current study's setting could be made clearer in the 'Context and Design' section. We have made several changes in the manuscript:

Under the "Context and design" section:

"This study was set up within the framework of a digital self-care development project initiated in 2018 and funded by the Swedish Ministry of Health and Social Affairs. The primary goal of this project is to improve the accessibility of psychological treatment in Sweden by creating concise self-care interventions derived from existing comprehensive digital interventions. The prospective single-group feasibility study was conducted at the Centre for Psychiatry Research, Stockholm Health Care Services, and Karolinska Institutet, Sweden. The study employed nationwide recruitment through advertisements on social media platforms."

And in the "discussion" section:

"Additionally, the recruitment methods differed between the reference study and the current study. The reference study used both advertisements and recruitment from healthcare professionals, while the current study solely relied on advertisements. This variation may have caused sample disparities. However, only 31 out of 132 participants in the reference study were recruited from a medical clinic. Further, the period between the two studies included a pandemic, potentially influencing the population's mental health awareness and levels of worry. This situation might have attracted different

types of patients interested in participating in an intervention focused on health anxiety. These factors render caution in all interpretation of differences and similarities in the comparison of interventions. Nevertheless, a strength lies in the fact that all participants in both the original and current studies underwent the same thorough diagnostic assessment, the two studies use the same eligibility criteria as well as that the level of health anxiety measured with a well-validated measure appear to be similar”.

And in the Abstract under “settings”

“Conducted at Karolinska Institutet, a medical university in Sweden, with nationwide recruitment through online advertisements.

Reviewer 2 comment #6:

All of the outcome measures were mostly self-report and lacked clinical validation. However, self-report measures may be subject to some biases, such as response bias and recall bias, and may not reflect the actual clinical diagnosis or severity of the disorders. How can these limitations be addressed or mitigated in the future studies?

We respond:

We thank the reviewer for this comment. We acknowledge the general limitations associated with the use of self-reporting instruments, such as the reliance on cooperation and the reading ability of the respondents, which could introduce various biases. Additionally, brief self-report scales might not encompass the entire range of symptoms related to the target condition. When designing the study, it was a balancing act between incorporating more instruments to best answer the study's questions and not unnecessarily burdening patients. It was also crucial to choose instruments that, in the event of intervention implementation, could be used in practice. If implemented, an intervention like this, would need to use as brief an instrument as possible and mostly based on self-report instrument in order to decrease burden on a potential clinic. The primary symptom measure used in this study was the SHAI-14, which has good psychometric properties. The same applies to the diagnostic interviews used before and after interventions (the MINI and the HPDI). The additional instruments were chosen to enable comparison with the original study. Regarding measure of adherence, this was based on both self-report as well as a more objective measure in terms of the number of completed modules. A potential improvement in a future study could be to conduct diagnostic interviews at post-treatment by a blind assessor. It would also be helpful to find a method for measuring the quantity of exposure exercises undertaken that does not rely solely on participants' self-reports. Finally, it would also be interesting to examine other aspects of improvement apart from changes in symptoms, such as healthcare consumption measured through patient registers.

In order to address the reviewer's comments we have added information regarding the diagnostic interview under the “measures” section:

“The HPDI can be administered with high interrater reliability (1)

And under limitations in the “discussion” section

“The current study primarily employed self-report measures. It is important to bear in mind the limitations inherent in these types of instruments, such as the reliance on the cooperation of the person answering them and the potential presence of recall and response bias. A potential improvement in a future study could be to conduct diagnostic interviews post-treatment by a blind assessor and/or measure changes in healthcare consumption before and after treatment, based on patient register data.

Reference:

1. Axelsson E, Andersson E, Ljotsson B, Wallhed Finn D, Hedman E. The health preoccupation diagnostic interview: inter-rater reliability of a structured interview for diagnostic assessment of DSM-5 somatic symptom disorder and illness anxiety disorder. *Cognitive behaviour therapy*. 2016;45(4):259-69.

Reviewer 3 comment #1: This is an informative case report which provides useful information about self-administered CBT computer interventions.

We respond:

We thank the reviewer for this positive feedback.

Reviewer 3 comment #2:

The methods suggest that the sample size was chosen to provide adequate precision in outcome measure estimates, but this is not defined quantitatively, e.g. a declaration about the widths of expected confidence intervals.

We respond:

We agree that this information should have been defined quantitatively and have added information regarding the rationale behind the choice of sample size under the "Participants and recruitment" section:

"For evaluation of the feasibility measures, a sample of 20 participants was deemed to be sufficient. Regarding preliminary within-group effect measured with the SHAI-14, to detect a moderate effect size (Hedges' $g = 0.6$), a sample of 24 participants was required, given 80% power and an α level set to $p < .05$ (2-tailed). To allow for some attrition, the goal was to recruit 25–30 participants."

Reviewer 3 comment #3:

Confidence intervals should be provided for binary outcome measures, e.g. remission.

We respond:

We thank the reviewer for this comment. We have now added confidence intervals for the variables "Below cut-off (remission) at 12 weeks" and "Below cut-off Remission at FU" in Table 2.

Reviewer 3 comment #4:

A list of the psychiatric comorbidities should be provided for the three groups of interest.

We respond:

We thank the reviewer for this suggestion. We agree that making this information more specific in the manuscript would be helpful. We have now expanded the 'comorbidity' section in Table 2 and included two subgroups for psychiatric comorbidity, namely depressive disorders and anxiety disorders. We have also clarified which diagnoses are included in these two groups in the table text:

"Psychiatric comorbidity refers to having a current DSM-5 diagnosis according to the MINI (17). "Depressive disorder" refers to major depressive disorder. "Anxiety disorders" refers to panic disorder, agoraphobia without panic disorder, social anxiety disorder, obsessive compulsive disorder, generalized anxiety disorder and post-traumatic stress disorder"

VERSION 2 – REVIEW

REVIEWER	Miller, J Washington University, Biostatistics
REVIEW RETURNED	30-Nov-2023
GENERAL COMMENTS	I would like to express my sincere appreciation for your prompt

	attention to my suggestions and comments on the manuscript. I have carefully reviewed the revised version, and I am pleased to acknowledge the significant improvements made. I commend the authors for addressing each of the points raised during the review process. The changes made to the manuscript have undoubtedly strengthened its overall quality. I particularly appreciate the effort put into refining the title, as the revised version now presents a clearer and more appropriate representation of the content. Moreover, the rewrite concerning patient involvement is noteworthy. The inclusion of more information on the patients' role enhances the manuscript's transparency and provides valuable insights into the importance of patient engagement in the study.
--	---